# Hypergraph-Based Machine Learning for Robust Handling of Missing Data

## Abstract

Handling missing data is a major challenge in machine learning where missing values are common in various datasets. This work introduces a hypergraph representation directly constructed from datasets containing missing values. The method does not rely on traditional techniques like deletion or data imputations. A hypergraph is directly extracted from the dataset, preserving the relationships between variables and modeling multi-variable interactions. This enables the model to capture the dataset structure in ways other methods may overlook. The proposed hypergraph learning method can be applied to classification and regression tasks. For real-world evaluation, we use the MIMIC-III and Adult datasets focusing on classification performance. Additionally, synthetic datasets with controlled missingness levels are used to evaluate the method's effectiveness across degrees of missing data. When compared with imputation and prediction techniques, the hypergraph approach achieves competitive or superior performance. Specifically, our method maintains high performance in scenarios with significant levels of missing data. We demonstrate that the hypergraph representation not only offers a more resilient framework for learning from datasets containing missing data. But also scales effectively across diverse datasets and prediction tasks. The method maintains stable performance under various degrees of missingness, demonstrating its potential as a valuable machine learning tool with high data reliability and prediction quality.

## 1 Introduction

Missing data in machine learning datasets is a major issue caused by various causes, such as human error, data corruption, or refusing to answer, leading to incomplete datasets. Groves et al. (2011), Yan et al. (2009) and Shih (2002) categorize missing data into three types: Missing Completely at Random (MCAR), Missing at Random (MAR), and Missing Not at Random (MNAR). These causes impact machine learning models in different ways. MCAR occurs when the missingness is unrelated to the data itself, MAR arises when the missingness is related to observed variables but not the missing ones, and MNAR occurs when the missingness is related to the missing data itself, leading to potentially severe biases Allison (2009).

Various approaches have been developed to handle missing data, a common issue across many fields. These methods aim to preserve the validity of statistical analyses and model predictions despite incomplete datasets. One of the most straightforward techniques is deletion, where instances with missing values are removed Little & Rubin (2019). Deletion methods, such as listwise or pairwise deletion, are the most straightforward approach but often significantly reduce sample size and increase potential bias if the data are not missing completely at random (MCAR) Little & Rubin (2020). This method simplifies data preparation, particularly when the proportion of missing data is low and the dataset is large. However, deletion methods have limitations, as they can introduce bias, reduce the data representativeness and distort relationships between variables. They lead to reduced statistical power and unreliable results Graham (2009).

More sophisticated approaches were introduced to address these issues. The imputation methods replace missing data with substituted values. The benefits of using imputation methods to handle missing data include the ability to retain valuable information by estimating missing values. Single imputation techniques, such as mean substitution, are easy to implement but fail to account for

some uncertainties. Multiple imputation addresses this issue by creating several imputed datasets and combining results across these datasets. Therefore, they can provide more reliable estimates and valid statistical inferences. Model-based methods, including maximum likelihood (ML) and Bayesian methods, offer a more rigorous approach by modeling the data with the missing values, assuming a particular distribution for the incomplete data Daniels & Hogan (2008).

Even though data imputations were commonly used, they present several problems. First, it may introduce bias. As the imputed values are based on statistical or prediction models, they may not fully capture the nature of the missing data. Especially when the data is not missing at random Little & Rubin (2019), Eekhout et al. (2012), and Collins et al. (2001). Imputation can reduce variability in data. When simple imputation techniques are used, they may lead to lower deviations and reduced variance Scheffer (2002). Imputation may fail to model relationships between variables, resulting in misleading and flawed interpretations Resche-Rigon & White (2018), and Kang (2013). Moreover, data imputation can foster overconfidence. When large portions of the dataset are missing. As the imputed data are treated as complete, prediction models may lead to wrong conclusions Van Buuren (2018), Dong & Peng (2013).

Knowledge graphs and hypergraphs are utilized to model complex relationships in datasets, providing advanced frameworks for representing simple and higher-order interactions. A knowledge graph represents entities as nodes and relationships between them as edges. It is widely adopted in domains such as natural language processing, recommendation systems, and the semantic web Paulheim (2017). Although highly effective for pairwise relationships, knowledge graphs often fall short in representing more complex interactions involving multiple entities, as they are restricted to binary connections between nodes Ji et al. (2021). To overcome this limitation, hypergraphs extend traditional graph models by introducing hyperedges, which can connect more than two nodes at once. Therefore capturing higher-order relationships within the data Zhou et al. (2006). This makes hypergraphs particularly useful in domains like bioinformatics and social networks where interactions often involve more than two elements Yadati et al. (2019). For example, in biological networks, a hypergraph can effectively represent complex interactions between multiple genes, proteins, or metabolic pathways, providing a richer model than the simple pairwise interactions Ahn et al. (2010).

Hypergraphs have also been successfully applied in machine learning tasks such as classification and clustering. In multi-label classification, for instance, hypergraph-based methods outperform traditional graph-based approaches by leveraging the multi-way relationships among labels and features Sun et al. (2008). Moreover, hypergraphs are proving to be particularly useful for handling missing data, where traditional methods struggle to capture the underlying structure of incomplete datasets Gao et al. (2020), Liu et al. (2017). By leveraging hypergraphs, researchers can uncover more intricate relationships in datasets and provide more accurate and robust analyses than knowledge graphs alone.

The previous researches in handling missing data primarily revolves around the complex relationship between features. While existing methods, such as imputation techniques and machine learning algorithms, have shown promise in addressing individual missing data, they often fall short of accurately capturing the intricate interdependencies. Moreover, most current techniques do not adequately address the variability introduced by missing data, potentially leading to biased estimates and conclusions. There is a need for more sophisticated models that can learn from incomplete data without making strong assumptions about the data mechanism or losing information introduced by missingness. Developing methods that can better understand and utilize the complex relationships between features in the presence of missing data remains a significant challenge in the field.

The structure of this work is organized as follows: Section 2 presents the proposed method. This section details how the hypergraph is constructed, and explains how the hypergraph can be used for inference. Section 3 describes the experimental setup, providing details about the datasets. And outlining the test to compare approaches. In Section 4, the experimental results are presented, and the proposed approach will be compared to other methods. Finally, Section 5 concludes the paper by summarizing the findings and suggesting potential directions for future work.

## 2 METHOD

To learn from datasets containing missing values, we propose a hypergraph representation that is able to capture complex relationships within these datasets. Hypergraphs are a generalization of graphs, consisting of nodes and hyperedges. A hyperedge is a subset of nodes. In this context, each node corresponds to a feature in the dataset. a hypergraph is used to represent variables and their interactions. Unlike traditional graphs where edges connect pairs of nodes, hypergraphs allow for hyperedges that connect multiple vertices simultaneously. They represent higher-order relationships among variables. This enables them to capture intricate interactions involving more than just two variables. Figure 1 shows an example of a hypergraph that displays relationships between diseases and conditions.

| | Hypertension | Lung cancer | Smoking | Cardio vascular | Diabetes | Obesity | Chronic kidney | Stroke | Liver disease |
|---|---|---|---|---|---|---|---|---|---|
| Hypertension | | 0.7 | 0.8 | | | | | | |
| Lung cancer | 0.7 | | 0.6 | | | | | | |
| Smoking | 0.8 | 0.6 | | 0.5 | | | | | |
| Cardio vascular | | | 0.5 | | | 0.4 | 0.9 | 0.2 | 0.5 |
| Diabetes | | | | | | 0.3 | | | |
| Obesity | | | | 0.4 | 0.3 | | 0.6 | | |
| Chronic kidney | | | | 0.9 | | 0.6 | | 0.5 | |
| Stroke | | | | 0.2 | | | 0.5 | | 0.7 |
| Liver disease | | | | 0.5 | | | | 0.7 | |

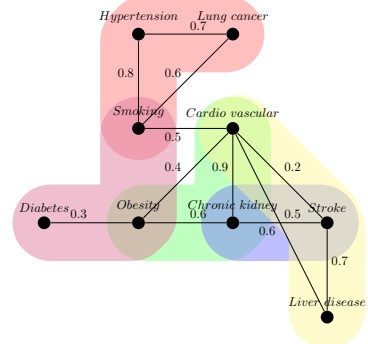

Figure 1: Hypergraph example

### 2.1 HYPERGRAPH CONSTRUCTION

Given a dataset, each node in the hypergraph corresponds to a feature or variable. The hyperedges represent relationships or interactions between subsets of nodes, allowing for more complex connections compared to traditional graph structures. Hypergraphs are particularly useful for representing variables and their interactions in scenarios where relationships involve more than two entities simultaneously. Formally, a hypergraph is defined as:

$$H = (V, E)$$

where:

- $V$ is a set of nodes (or vertices), i.e., $V = \{v_1, v_2, \ldots, v_n\}$.
- $E$ is a set of hyperedges, where each hyperedge is a subset of $V$, i.e., $e_i \subseteq V$ for each $e_i \in E$.

In other words, each hyperedge $e_i$ can connect multiple vertices simultaneously, which distinguishes hypergraphs from traditional graphs. The degree of a vertex $v \in V$ is the number of hyperedges that contain the vertex.

Each hyperedge represents a subset of dataset samples that share the same number of available parameters. This facilitates the organization of complete and incomplete data, allowing the grouping of samples with similar patterns of missing data. To ensures that subgraphs derived from larger sample sizes have greater influence on the analysis, each hyperedge $e_i \in E$ is weighted by the number of samples $N(e_i)$ that contribute to it. The weight of hyperedge $e_i$ is defined as:

$$w(e_i) = N(e_i)$$

For each pair of variables within a hyperedge, we define edges that represent the regression relationships between the variable pairs. The weight of each edge is determined by the correlation coefficient $\rho_{ij}$ between two variables $v_i$ and $v_j$. The edge weight is defined as:

$$w(v_i, v_j) = \rho_{ij}$$

where $\rho_{ij}$ is the Pearson correlation coefficient calculated between variables $v_i$ and $v_j$. This weight reflects the strength of the relationship between the variables.

As summary, to generate the hypergraph dataset samples are grouped based on identical patterns of missing variables. Each distinct missingness pattern forms a unique hyperedge $e_i$, and the weight of each hyperedge is assigned according to the number of samples contributing to its construction. Thus, each hyperedge weight $w(e_i)$ represents the prevalence of a specific missingness pattern, while the pairwise relationships between variables within each hyperedge are weighted by the correlation coefficient $\rho_{ij}$, providing a quantitative measure of the relationship strength.

## 2.2 HYPERGRAPH INFERENCE

The hypergraph representation effectively encodes the interdependencies between variables and the missingness characteristics within a dataset. This encoding allows the hypergraph to generalize observed data and infer values for missing or unknown variables. The inference process for predicting unseen samples using the hypergraph can be outlined as follows:

To make predictions, we first identify the hyperedges that contain the unknown target variable. For each selected hyperedge $e_i \in E$, the values of its vertices (nodes) are assigned based on the values from the sample for which we are making the prediction. Let the sample values be denoted as $\mathbf{x} = \{x_1, x_2, \ldots, x_n\}$, where each $x_i$ corresponds to a value of vertex $v_i$.

Once the values from the sample are assigned to the vertices, we traverse the hypergraph from vertices that have known values to predict the values of their neighboring vertices (nodes with missing values). The predicted value of a neighboring vertex $v_j$ is calculated using the regression coefficients $\beta_{ij}$, which were derived during hypergraph generation, as follows:

$$\hat{x}_j = \sum_{v_i \in \text{neighbors of } v_j} \beta_{ij} x_i$$

where $\hat{x}_j$ is the predicted value of vertex $v_j$, and $x_i$ are the known values of its neighboring vertices. The regression coefficient $\beta_{ij}$ reflects the relationship strength between vertices $v_i$ and $v_j$.

If the target vertex $v_j$ has edges connecting it to multiple neighboring vertices, the final predicted value is computed as a weighted average of the predictions from these neighbors. The weights are determined by the correlation coefficients $\rho_{ij}$ associated with the edges connecting $v_j$ to its neighbors:

$$\hat{x}_j = \frac{\sum_{v_i \in \text{neighbors of } v_j} \rho_{ij} \hat{x}_i}{\sum_{v_i \in \text{neighbors of } v_j} \rho_{ij}}$$

where $\hat{x}_i$ represents the predicted values from each neighbor, and $\rho_{ij}$ is the correlation coefficient between variables $v_i$ and $v_j$.

The final predicted value $\hat{y}$ for the unknown target variable is obtained by integrating the predictions from multiple hyperedges. A weighted average of the predictions is taken, where the weights $w(e_i)$ are derived from the hyperedge weights, reflecting the significance of each hyperedge:

$$\hat{y} = \frac{\sum_{e_i \in \mathcal{H}} w(e_i) \hat{x}_j(e_i)}{\sum_{e_i \in \mathcal{H}} w(e_i)}$$

where $\hat{x}_j(e_i)$ is the predicted value from hyperedge $e_i$, and $w(e_i)$ is the weight of hyperedge $e_i$, representing the number of samples contributing to that hyperedge.

## 3 EXPERIMENT

We will evaluate our method on real and synthetic datasets to assess its performance across different scenarios. The real datasets used in this study include MIMIC-III and Adult datasets which naturally contain some missing values, providing a realistic testbed for imputation and prediction tasks. To further explore how our method handles varying degrees of missing data, we will generate synthetic datasets with controlled missingness rates, ranging from 0% to 60%. These synthetic datasets will allow us to systematically study the impact of different levels of missing data on model performance and ensure robustness across a wide range of conditions.

### 3.1 REAL DATASET

#### 3.1.1 MIMIC-III

The MIMIC-III (Medical Information Mart for Intensive Care III) dataset is a large, publicly available database comprising de-identified health data from over 40,000 critical care patients admitted to the Beth Israel Deaconess Medical Center between 2001 and 2012 Johnson et al. (2016). It includes detailed information such as patient demographics, vital signs, laboratory results, medications, procedures, diagnostic codes, and clinical notes. The dataset is widely used for medical research, particularly in predictive modeling, due to its richness in temporal and multimodal data. It allows researchers to develop and validate machine learning models in a healthcare context.

To preprocess the MIMIC-III dataset for our experiments, we selected features that included demographic information, patient monitoring data, and laboratory results. Continuous variables were normalized to ensure consistent scaling across all features, and categorical variables were one-hot encoded to convert them into a numerical format suitable for machine learning models. Finally, the dataset was split into training and testing sets in a 4 to 1 ratio. For our experiment, we perform classification task of predicting whether a patient's length of stay (LOS) will exceed 3 days in the ICU. This is a common challenge using the MIMIC-III dataset. The complexity of ICU patients and the dynamic nature of their conditions make this task challenging, requiring sophisticated models.

#### 3.1.2 ADULT

The Adult dataset, also known as the "Census Income" dataset, is a widely used dataset in machine learning research and comes from the 1994 U.S. Census Bureau data. It contains demographic and employment-related attributes for individuals such as age, education level, occupation, marital status, work hours per week, and native country. The dataset has over 48,000 records with categorical and numerical features, and includes some missing values. Its diversity and real-world nature make it a popular choice for classification tasks.

The prediction task associated with the Adult dataset is to determine whether an individual earns more than $50K a year based on their demographic and employment features. This binary classification problem involves using attributes to predict income level. The task is a benchmark problem for testing classification algorithms, as it requires handling a mix of categorical and continuous data, missing values, and potential biases in the dataset.

### 3.2 SYNTHETIC DATASET

The second test is performed on synthetic datasets designed with varying degrees of missingness. These are critical to assessing the versatility and resilience of our approach. A controlled environment allows us to systematically evaluate the robustness and accuracy of our method under different conditions. In this test, we focus on regression tasks to measure the method's ability to accurately predict continuous outcomes despite missing data and complex variable interactions. We evaluated the performance against several imputation and regression methods. Specifically, logistic regression, support vector regression, gaussian process, random forest and multi layer perceptron were tested. These models were chosen to assess our approach's effectiveness against different regression algorithms.

For synthetic datasets, we designed the target variable to exhibit a correlation coefficient with other variables in the range of 0.6 to 0.9, ensuring a meaningful relationship between the target and pre-

Table 1: Experiment results on real datasets

| DATASET | CLASSIFICATION | ACCURACY (%) | | | | |
|---|---|---|---|---|---|---|
| | | | IMPUTATION | | | |
| | | - | Deletion | Mean | NN | MICE |
| MIMIC-III (LOS>3 Days) | Hypergraph (Proposed) | **75.7** | - | - | - | - |
| | Support Vector | - | 65.0 | 67.2 | 69.3 | 70.8 |
| | Gaussian Process | - | 62.7 | 65.3 | 67.2 | 69.5 |
| | Decision Tree | - | 66.8 | 69.6 | 69.0 | 71.6 |
| | Random Forest | - | 67.5 | 67.2 | 72.2 | 74.1 |
| | Multi Layer Perceptron | - | 68.3 | 70.1 | 71.1 | 75.1 |
| Adult (Income> $50,000 a year) | Hypergraph (Proposed) | 85.8 | - | - | - | - |
| | Support Vector | - | 72.7 | 76.0 | 85.6 | 85.6 |
| | Gaussian Process | - | 77.1 | 77.6 | 80.2 | 82.3 |
| | Decision Tree | - | 72.4 | 75.9 | 78.2 | 81.0 |
| | Random Forest | - | 78.3 | 78.1 | 79.7 | 86.9 |
| | Multi Layer Perceptron | - | 75.4 | 74.2 | 84.9 | **87.2** |

dictor variables. The datasets consisted of 10 predictor variables and 1 target variable, with a total of 10,000 samples to provide sufficient data for a robust evaluation. The data was split into a 4 to 1 ratio for training and testing, respectively, allowing us to evaluate the model's generalization on unseen data. Root Mean Squared Error (RMSE) was used as the primary evaluation metric to assess model performance. This experiment allowed us to systematically examine the behavior and efficacy of the proposed hypergraph method across different controlled levels of missingness.

## 4 EXPERIMENTAL RESULTS

In this section, we present the results of our proposed hypergraph-based method in comparison to traditional machine learning approaches that rely on data imputation techniques followed by standard classifiers. Specifically, we evaluated the performance of several widely used imputation methods, including deletion, mean, nearest neighbors and MICE imputation Van Buuren & Oudshoorn (2000), coupled with classifiers and regression models. The first result group is the test on real datasets as detailed in the the previous section. Then, the result of the experiment on synthetic dataset.

Table 1 presents the experimental results on real-world datasets. Demonstrating that the proposed hypergraph-based method achieves comparable or higher values compared to traditional imputation and classification methods. For MIMIC-III dataset classifying length of stay, the proposed method performs slightly better than the traditional techniques which rely on imputation methods. For Adult dataset, the proposed method performs in a similar level to the best results of traditional methods.

Table 2 presents the experimental results on synthetic datasets, with the leftmost column listing specific missingness levels ranging from 0% to 60%. For the case of no missing data (0% missingness), no data imputation was performed, and all methods were evaluated directly. The table shows the RMSE (Root Mean Squared Error) for each missingness level, comparing the performance of our proposed hypergraph-based method with traditional imputation and regression techniques. The overall results indicate that our method performs particularly well as the missingness increases. Notably at 40% and 60% missingness, the hypergraph approach consistently outperforms the traditional imputation and regression methods, demonstrating its robustness and effectiveness in handling substantial levels of missing data. These findings highlight the resilience of the hypergraph method in maintaining prediction accuracy even in challenging scenarios with high degrees of missingness.

Table 2: Experiment results on synthetic datasets

| | | RMSE | | | | |
| | | | IMPUTATION | | | |
| MISSING (%) | REGRESSION | - | Deletion | Mean | NN | MICE |
|---|---|---|---|---|---|---|
| 0 | Hypergraph (Proposed) | 43.9 | - | - | - | - |
| | Logistic Regression | 65.0 | - | - | - | - |
| | Support Vector | 61.8 | - | - | - | - |
| | Gaussian Process | 42.7 | - | - | - | - |
| | Random Forest | **40.1** | - | - | - | - |
| | Multi Layer Perceptron | 45.5 | - | - | - | - |
| 20 | Hypergraph (Proposed) | 45.4 | - | - | - | - |
| | Logistic Regression | - | 65.6 | 68.0 | 62.4 | 64.1 |
| | Support Vector | - | 63.4 | 65.6 | 61.7 | 63.0 |
| | Gaussian Process | - | 48.1 | 51.7 | 47.9 | 49.3 |
| | Random Forest | - | 47.0 | 47.5 | **45.1** | 46.8 |
| | Multi Layer Perceptron | - | 49.9 | 50.9 | 48.4 | 49.1 |
| 40 | Hypergraph (Proposed) | **66.9** | - | - | - | - |
| | Logistic Regression | - | 98.1 | 100.3 | 93.8 | 96.4 |
| | Support Vector | - | 90.5 | 92.1 | 88.6 | 90.9 |
| | Gaussian Process | - | 83.4 | 85.6 | 82.2 | 85.8 |
| | Random Forest | - | 70.6 | 71.1 | 67.8 | 69.2 |
| | Multi Layer Perceptron | - | 75.4 | 75.6 | 74.2 | 75.4 |
| 60 | Hypergraph (Proposed) | **95.4** | - | - | - | - |
| | Logistic Regression | - | 130.9 | 135.1 | 120.1 | 130.8 |
| | Support Vector | - | 127.6 | 130.7 | 128.5 | 127.6 |
| | Gaussian Process | - | 120.2 | 117.0 | 125.4 | 119.0 |
| | Random Forest | - | 117.8 | 109.0 | 105.6 | 99.8 |
| | Multi Layer Perceptron | - | 120.3 | 110.2 | 103.4 | 105.6 |

## 5 CONCLUSION

This paper presents a novel hypergraph-based approach for handling missing data in machine learning, offering a robust alternative to traditional methods like deletion or imputation. Directly constructing hypergraphs from datasets, effectively preserves variable relationships and models multi-variable interactions, leading to improved performance in classification and regression tasks. Through comprehensive evaluations on real-world datasets as well as synthetic datasets with controlled missingness, the proposed method demonstrates highly competitive performance compared to other methods. It consistently achieves accuracy that are comparable to, or better than, those obtained using imputation techniques and traditional classifiers. Notably, the hypergraph representation excels in scenarios with substantial missing data. Furthermore, a notable advantage of our hypergraph-based method is its consistency. The results show that it provides reliable performance across all datasets with varying missingness levels. This consistency is crucial for real-world applications.

Beyond its demonstrated effectiveness in handling missing data within individual datasets, the proposed hypergraph-based method shows great potential for cross-dataset learning, particularly in scenarios where feature sets differ significantly between datasets. In such cases, merging datasets often results in substantial amounts of missing data due to the absence of overlapping features. Traditional approaches to address this issue, such as imputation or deletion, can lead to information loss or introduce bias. However, the hypergraph representation naturally accommodates missing values while preserving the multi-variable relationships inherent to each dataset. This allows the model to

leverage the complementary information present in different datasets without relying on imputation, making it highly suited for cross-dataset applications. As a result, this method opens up promising avenues for combining heterogeneous datasets in fields such as healthcare, where multiple data sources often produce similar datasets.

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
