# OpenReview forum: "Hypergraph-Based Machine Learning for Robust Handling of Missing Data"
_ICLR.cc/2025/Conference — ICLR 2025 Conference Withdrawn Submission_

### Official Review · Reviewer_8RRS · 2024-10-28

**Soundness:** 1
**Presentation:** 1
**Contribution:** 1
**Rating:** 1
**Confidence:** 4

**Summary:**

This paper presents a method to handle missing values by constructing a hypergraph.

**Strengths:**

- The author introduce what is a hypergraph and show an example (without missing values)

**Weaknesses:**

- The methodology is unclear for me and the presentation is poor quality.
See Questions for more details.

**Questions:**

Methodology (only few questions):
- How the Pearson correlation coefficient is calculated ? Only on observed samples ?
If yes, the method can only handle MCAR data and it must be notified. In addition, if the percentage of missing values is high, the Pearson correlation could be computed on very few samples.
- I really do not understand how the hypergraph is constructed in the presence of missing data. "As summary, to generate the hypergraph dataset samples are grouped based on identical patterns of missing variables. Each distinct missingness pattern forms a unique hyperedge $e_i$, and the weight of each hyperedge is assigned according to the number of samples contributing to its construction." Maybe the authors could take an example.
- I also do not understand how the regression coefficients $\beta$ are computed. Only on observed samples ?

Experimental study:
- The authors introduce the missing-data mechanisms but do not mention it in the experimental study. How have missing data been simulated ?
- The authors compare their method with the mean imputation, KNN imputation and MICE. Line 307 some references are not given. The method should be compared with other imputation methods, like missForest which is widely used.
Stekhoven, Daniel J., and Peter Bühlmann. "MissForest—non-parametric missing value imputation for mixed-type data." Bioinformatics 28.1 (2012): 112-118.

Presentation:
As it is, it's difficult to read the article.
- Some sentences are unfinished: for example line 66: When large portions of the dataset are missing.
- The authors can use \citep and \citet with the natbib package in LaTex to cite the works in a sentence. For exemple: "when the missingness is related to the missing data itself, leading to potentially severe biases **[Allison,2009]**". And not "when the missingness is related to the missing data itself, leading to potentially severe biases Allison (2009)"

---

### Official Review · Reviewer_aeis · 2024-10-31

**Soundness:** 2
**Presentation:** 1
**Contribution:** 2
**Rating:** 3
**Confidence:** 5

**Summary:**

This paper proposes a hypergraph-based machine learning method to address the common issue of missing data in datasets. Unlike traditional deletion and imputation methods, this approach constructs a hypergraph directly from the original dataset, where nodes represent data features, hyperedges represent samples with the same missing patterns, and weights reflect the sample size and the strength of variable relationships. The constructed hypergraph is then used to learn from the missing data through a process of hypergraph inference.

**Strengths:**

The method proposed in this paper, which uses hypergraph inference to learn from missing data, shows a degree of innovation. Missing data is a challenging problem in machine learning, particularly in high-value fields such as healthcare. This approach holds considerable reference value for research in related areas.

**Weaknesses:**

1. The primary innovation of this paper lies in using hypergraphs to learn from missing data. However, the data inference section does not fully demonstrate the advantages of hypergraph-based data aggregation and instead relies on calculation methods similar to those used in traditional graphs. In the field of missing data, there are existing methods for constructing graph structures, but this paper does not reference these methods, nor does it highlight the advantages of hypergraph construction over traditional graph structures.

2. The paper defines hyperedges as collections of samples with the same missing patterns, but it lacks sufficient explanation and examples of these patterns. In the hypergraph inference section, the calculations for single-neighbor nodes and multi-neighbor nodes are expressed through two formulas, which essentially represent the same algorithm, making it overly redundant.

3. The paper uses the Pearson correlation coefficient to measure relationships between nodes but lacks an explanation of this metric and the rationale for its selection.

**Questions:**

1. Advantages of using hypergraph structures over graph structures.

2. Specific explanation and examples of missing patterns.

---

### Official Review · Reviewer_mTNM · 2024-11-01

**Soundness:** 1
**Presentation:** 2
**Contribution:** 1
**Rating:** 3
**Confidence:** 2

**Summary:**

The paper proposes a method to impute missing data based on hypergraphs, which are subsets of the features that are used to predict the missing values of the features within the subsets.

**Strengths:**

- I think the method exploit relationships among features in a specific way to predict missing data.

**Weaknesses:**

- The main problem of the paper is that the problem of previous work and the motivation of the work are not explained in the method to a coherent level.
- There is no analysis of all previous methods to identify the problem for this paper. Interdependency among many features are also well studied it is not clear what is the difference this paper can make.
- There is no method idea section in intro, nor in the method section that can help to introduce high-level rationale of the the method.
- In the method, the construction of hyperedges is not clearly described.
- Overall, the method is not clearly described. The rationale and the strength of the method compared to previous method is missing.

**Questions:**

- While there are many method that do not use interdependency among features, there are many using the interdependency. What are all of them? What are they all missing that is addressed in this paper?
- It is cited in the paper that hypergraphs are particularly useful for missing data before (Gao 2020, Liu 2017), so what is contribution of the the current paper on top of these related methods?
- Here is what written that are highly confusing: "each node in the hypergraph corresponds to a feature or variable." and "Each hyperedge represents a subset of dataset samples that share the same number of available parameters". So, the hyperedges are subgroup of features or samples?

---

### Official Review · Reviewer_95R2 · 2024-11-04

**Soundness:** 2
**Presentation:** 2
**Contribution:** 2
**Rating:** 3
**Confidence:** 4

**Summary:**

This paper presents a hypergraph-based method to address missing data. The authors claim that hypergraphs can be directly constructed from incomplete data while modeling complex interconnections between variables.

**Strengths:**

- The proposed method uses hypergraphs to capture higher-order interconnections between variables, which is a novel approach in addressing missing data.
- The use of realistic datasets, such as MIMIC-III, distinguishes this study from existing studies that primarily relies on benchmark datasets from the UCI repository.

**Weaknesses:**

- Lack of logical structure in the introduction: The introduction fails to effectively highlight the goal of the research. While the motivations for the study are present, they are scattered throughout the section

- Insufficient comparison with recent studies: The paper lacks a dedicated related work section. While a related work section is not always necessary if it was already integrated into the introduction, I don't think that is the case here. Also, the studies mentioned are outdated. The authors should include more recent and sophisticated methods for missing data imputation in both the literature review and experiments.

- Unclear description of the proposed method: I am sorry. I am not particularly familiar with hypergraphs, but I think the explanation of hypergraph construction and inference is insufficient. Without a clear understanding of the methodology, it was challenging to assess the paper's true contribution. Moreover, I think there might be a discrepancy between the motivation and the proposed method. The authors argue that previous research is limited in the use of pairwise connectivity, but the proposed method also seems to use pairwise connections in its hypergraph structure. The only difference I noted is that the proposed method excludes irrelevant features, but I don't think this is an inherent property of hypergraphs.

- Limited experiments: Although the authors attempted to demonstrate practicality by using realistic datasets like MIMIC-III, the evaluation lacks diversity. The authors should include more benchmark datasets, as is typical in missing data imputation research, to demonstrate the method's superiority. Additionally, authors can simulate missing data in realistic datasets by intentionally removing values to test the method's robustness. I think the generated synthetic data is somewhat ideal in that features are too relevant so it is not appropriate to assess true robustness, particularly for higher missing rates.

**Questions:**

1. Could you provide more details on how the hypergraph is constructed and how it helps capture higher-order relationships beyond pairwise interactions? This would help highlight the method's strengths more effectively.

2. Why did you choose simple prediction methods with hypergraphs? I think this simplicity can be a bottleneck for the proposed method. I would like to understand why you concluded that these simple prediction methods are enough.

3. As I know, MIMIC-III is not highly structured. If so, it may contain natural missingness, and imputing these missing values is undesirable. With other imputation-based methods, distinguishing between natural and unintended missingness is challenging. Is your hypergraph-based method particularly helpful in such situation?

---

### Note · Authors · 2024-11-28

I have read and agree with the venue's withdrawal policy on behalf of myself and my co-authors.